# A Six-Week Follow-Up Study on the Sustained Effects of Prolonged Water-Only Fasting and Refeeding on Markers of Cardiometabolic Risk

**DOI:** 10.3390/nu14204313

**Published:** 2022-10-15

**Authors:** Sahmla Gabriel, Mackson Ncube, Evelyn Zeiler, Natasha Thompson, Micaela C. Karlsen, David M. Goldman, Zrinka Glavas, Andrew Beauchesne, Eugene Scharf, Alan C. Goldhamer, Toshia R. Myers

**Affiliations:** 1TrueNorth Health Foundation, Santa Rosa, CA 95404, USA; 2Department of Research, American College of Lifestyle Medicine, Chesterfield, MO 63006, USA; 3MetaBite, Dover, DE 19901, USA; 4School of Medicine, Tufts University School of Medicine, Boston, MA 02111, USA; 5Department of Neurology, Mayo Clinic, Rochester, MN 55905, USA; 6TrueNorth Health Center, Santa Rosa, CA 95404, USA

**Keywords:** prolonged fasting, water-only fasting, cardiometabolic health, insulin resistance, hypertension, hyperlipidemia, fatty liver index, whole-plant-food diet

## Abstract

(1) Background: Chronic inflammation and insulin resistance are associated with cardiometabolic diseases, such as cardiovascular disease, type 2 diabetes mellitus, and non-alcoholic fatty liver disease. Therapeutic water-only fasting and whole-plant-food refeeding was previously shown to improve markers of cardiometabolic risk and may be an effective preventative treatment but sustained outcomes are unknown. We conducted a single-arm, open-label, observational study with a six-week post-treatment follow-up visit to assess the effects of water-only fasting and refeeding on markers of cardiometabolic risk. (2) Methods: Patients who had voluntarily elected and were approved to complete a water-only fast were recruited from a single-center residential medical facility. The primary endpoint was to describe changes to Homeostatic Model Assessment of Insulin Resistance (HOMA-IR) scores between the end-of-refeed visit and the six-week follow-up visit. Additionally, we report on changes in anthropometric measures, blood lipids, high-sensitivity C-reactive protein (hsCRP), and fatty liver index (FLI). Observations were made at baseline, end-of-fast (EOF), end-of-refeed (EOR), and six-week follow-up (FU). (3) Results: The study enrolled 40 overweight/obese non-diabetic participants, of which 33 completed the full study protocol. Median fasting, refeeding, and follow-up lengths were 14, 6, and 45 days, respectively. At the FU visit, body weight (BW), body mass index (BMI), abdominal circumference (AC), systolic blood pressure (SBP), diastolic blood pressure (DBP), total cholesterol (TC), low-density lipoprotein (LDL), hsCRP, and FLI were significantly decreased from baseline. Triglycerides (TG) and HOMA-IR scores, which had increased at EOR, returned to baseline values at the FU visit. (4) Conclusion: Water-only fasting and whole-plant-food refeeding demonstrate potential for long-term improvements in markers of cardiovascular risk including BW, BMI, AC, SBP, DBP, blood lipids, FLI, and hsCRP.

## 1. Introduction

The burden of chronic metabolic diseases, such as cardiovascular disease (CVD), hypertension (HTN), type 2 diabetes mellitus (T2D), and non-alcoholic fatty liver disease (NAFLD) is a growing global public health concern because they may result in diminished quality of life, disability, and risk of premature death [1]. Obesity, primarily visceral fat accumulation, promotes a chronic pro-inflammatory state, which is linked to the development of insulin resistance—the primary step in metabolic disease pathogenesis [2]. Preventative and therapeutic interventions aimed at managing systemic inflammation and insulin resistance may lessen disease progression and improve quality of life [3].

Fasting is the partial or complete abstinence of caloric intake for a defined period of time. Research into intermittent fasting, the fasting mimicking diet, and prolonged fasting methods have demonstrated the potential of these therapies to improve overall health and promote immunity and longevity [4]. Prolonged fasting protocols have been shown to improve cardiometabolic markers associated with obesity such as insulin sensitivity, blood lipids, body weight, and abdominal circumference [5,6,7]. Prolonged fasting is typically conducted as a very-low-calorie or zero-calorie (e.g., water-only fasting) intervention for a period of 2 or more days [8]. The gradual reintroduction of food—typically for a period of at least half of the total fast length—is essential after a prolonged caloric deficit in order to properly account for any metabolic and electrolyte changes and prevent refeeding syndrome [8]. Although various refeeding diets have been reported [9,10], there is a lack of knowledge regarding the metabolic changes that occur during food reintroduction following prolonged water-only fasting.

We recently reported that at least 10 days of water-only fasting and five days of refeeding on an exclusively whole-plant-food diet free of added salt, oil, and sugar (SOS-Free Diet) had mixed effects on markers of cardiovascular and metabolic health at the end of refeed in a non-diabetic, overweight/obese population [6]. We observed clinically meaningful reductions in body weight (BW), abdominal circumference (AC), blood pressure (BP), total cholesterol (TC), low-density lipoprotein (LDL), and high-sensitivity C-reactive protein (hsCRP). However, increases in very-low-density lipoprotein (VLDL), triglycerides (TG), glucose, and insulin were also observed. Calculated homeostatic model of insulin resistance (HOMA-IR) scores, which reduced after fasting, had increased significantly after the refeeding period, indicating increased insulin resistance. We hypothesized that these effects may be a transient rebound phenomena observed during the refeeding period as fatty acid metabolism, activated during fasting, switches back to glucose metabolism during refeeding [6,11]. To further elucidate these findings, we conducted a follow-up, single-arm, open-label, observational study on the effects of water-only fasting and refeeding with an additional six-week post-treatment in a non-diabetic, overweight/obese population.

## 2. Materials and Methods

### 2.1. Ethical Statement

This study (NCT04514146) was conducted according to the guidelines of the Declaration of Helsinki, and approved by the Institutional Review Board of the TrueNorth Health Foundation (TNHF-2020-2VAT).

### 2.2. Participation and Visit Characteristics

We enrolled 40 non-diabetic, overweight and obese participants who had elected to undergo a medically supervised, water-only fast at a residential fasting facility prior to recruitment. Eligibility included males and females aged 40–70 years with a body mass index (BMI) between 25–40 kg/m^2^ and fasting glucose <7 mmol/L and/or hemoglobin A1c <7% who were approved by a physician to water-only fast for at least 10 consecutive days followed by the standard refeeding period of no less than half the duration of the fast. Exclusions were active malignancy, active inflammatory disorders including classic autoimmune connective tissue disorders, multiple sclerosis, inflammatory bowel disorders, and stroke or heart attack within the last 90 days. Participation continued until six-week post-treatment data collection was completed. Of the 40 eligible participants, 17 consenting participants were concurrently enrolled in a study assessing the safety and feasibility of prolonged water-only fasting in the treatment of stage I and II hypertension (NCT04515095).

Of the 40 participants enrolled, 38 completed the on-site fasting and refeeding treatment protocol and 33 completed the full study protocol including the six- week follow-up (FU) visit. Of the 33 who completed the full protocol, four were unable to return for the on-site FU visit and were provided the necessary materials and education to collect and report data remotely (i.e., at home). Two participants did not complete the minimally required fasting length due to treatment emergent adverse events including one grade 1 nausea event and one grade 2 dyspepsia event. There were two participants whose water-only fast was temporarily interrupted. In one case, the fast was interrupted after day four of water-only fasting with two days of vegetable and fruit juice due to one episode of emesis. In the other case, the fast was interrupted after day six of water-only fasting with one day of cooked zucchini/squash blend to alleviate symptoms of emesis. In both cases, water-only fasting resumed once symptoms subsided. All reported adverse events were mild (grade 1) to moderate (grade 2) events, consistent with previously reported water-only fasting safety data [8]. There were no severe or serious adverse events reported during fasting or refeeding. Common treatment emergent adverse events included grade 1 headache, fatigue, minor muscle cramp, dyspepsia, nausea, and vomiting. One participant with a pre-existing history of gout experienced gout symptoms during the fast. Two participants experienced low potassium and their original fasting plan was shortened, but they met the minimum fast length criteria and remained in the study.

### 2.3. Medically Supervised, Water-Only Fasting and Refeeding Protocol

The medically supervised, water-only fasting and refeeding protocol was implemented by non-research medical personnel at a residential, medical facility according to the facility’s standard protocol as previously reported [8]. Briefly, potential participants were pre-screened before arrival and, if conditionally approved to water-only fast, were instructed to eat a diet consisting of fresh fruits and raw or steamed vegetables for two days prior to initiating the fast. Participants were instructed to consume a minimum of 40 ounces of distilled water per day and limit physical activity during the course of the fast. While fasting, participants remained on-site and non-research medical personnel monitored vital signs and symptoms twice daily along with weekly serology and urine analysis to monitor electrolyte balance and other physiological functions, such as kidney and liver function. Adverse events were continuously monitored and if necessary the fast was temporarily interrupted with vegetable broth or juice or suspended by initiating the standard refeeding protocol. The fast was broken with a refeeding process consisting of five phases of gradual food introduction, with one phase for every 7–10 days of fasting. Phase one is a mixture of fruit and vegetable juice; phase two includes the addition of raw fruits and vegetables; phase three includes the addition of steamed vegetables; phase four includes the addition of whole grains; and phase five includes the addition of legumes, until participants are eating an exclusively SOS-Free Diet. Only phase one limited daily calorie consumption, and each new phase consisted of a continuation of items from the previous phase with the addition of more complex foods. Refeeding length was at least half of the fasting length. Additionally, participants’ vital signs and symptoms were monitored twice daily for the duration of the treatment, which included fasting and refeeding phases. Approved fasting lengths varied but were no less than 10 days followed by a refeeding period of at least five days.

### 2.4. Study Design

Eligible and consenting participants attended study visits at baseline, end of fast (EOF), end of refeed (EOR), and six-week FU. At each visit, 18 mL of blood were collected, and clinical measurements, which included resting systolic and diastolic blood pressure (SBP and DBP), BW, and AC, were taken. After collection, blood samples were sent to Labcorp for measurements of fasting blood glucose, insulin, hsCRP, blood lipids, and gamma-glutamyl-transferase (GGT). Measured values of glucose and insulin were used to calculate HOMA-IR score according to the equation (fasting insulin (microU/L) × fasting glucose (nmol/L)/22.5) [12]. Measured *BMI*, *AC*, *TG*, and *GGT* were used to calculate Fatty Liver Index (FLI) via the previously established formula [13]:$$FLI=\frac{\left(e^{0.953*log_{e}TG+0.139*BMI+0.718*log_{e}GGT+0.053*AC-15.745}\right)}{\left(1+e^{0.953*log_{e}TG+0.139*BMI+0.718*log_{e}GGT+0.053*AC-15.745}\right)\;}*100$$

Demographic information, such as age, sex, ethnicity, diet type, and pre-treatment ICD-10 diagnostic codes, was collected once at baseline. Participants also answered an online dietary screener survey at baseline and FU.

### 2.5. Clinical Measurements

Clinical measurements were obtained at baseline, EOF, EOR, and FU by research personnel. Height was only measured at baseline. Prior to any measurements, participants were instructed to remove shoes, coats, and any pocket items.

Height (cm) was measured using a digital wall-mounted stadiometer (DS5100, Doran Scales Inc., St. Charles, IL, USA) Participants were positioned with backs against the stadiometer, with legs, backs, neck, and head straight. Once in position, the head piece was lowered to touch participants’ crown of the head.

BW (kg) was measured using digital body weight scale (BWB 800A Class III, Tanita Corporation of American Inc., Arlington Heights, IL, USA). Participants stepped on the scale and stood still until the measurement was produced and recorded. The participants who elected to complete their FU visit remotely were provided a Conair digital glass scale (WW26 model). BMI was calculated from height and weight using the formula: weight (kg) ÷ height (m^2^).

AC (cm) was measured at the narrowest part of participants’ midsection below the lowest palpable rib cage yet above the top of the hip/pelvic bones. Tape was placed horizontally and measurements were collected using a tension-sensitive, non-elastic tape (Gullick II, Model 67019, Country Technology Inc., Gay Mills, WI, USA). Participants who elected to complete their FU visit remotely were provided a soft retractable cloth measuring tape.

Resting SBP and DBP (mmHg) were measured in the morning after blood collection. Participants rested for five minutes in the seated position and blood pressure was measured using a digital blood pressue device (Welch Allyn-Connex ProBP 3400, Hill-Rom Holding Inc. Chicago, IL, USA) Participants who completed their FU visit remotely were provided a digital blood pressure device with adjustable cuff size (BP3GX1, Microlife USA Inc, Clearwater, FL, USA).

### 2.6. Laboratory Measurements

Blood was collected in the seated position by a certified phlebotomist in the morning prior to any caloric food or drink consumption. Participants were instructed to drink 1–2 glasses of water prior to collection. Blood was drawn into vacutainer tubes (Red top, 16 × 100, 10 mL, silica, BD, Mississauga, ON, Canada). Red top tubes were incubated for 30 min at room temperature before being centrifuged at 1500× *g*, 10 min, 4 °C. The serum was separated, refrigerated at 4 °C, and samples were sent to a commercial lab (Labcorp, Burlington, NC, USA) for analysis [6]. LabCorp reports the following: Glucose was determined by enzymatic reference method with hexokinase and UV test; Insulin was determined by electrochemiluminescence immunoassay; hsCRP was determined via particle enhanced immunoturbidimetric assay, where human CRP agglutinates with latex particles were coated with monoclonal anti-CRP antibodies and precipitate was determined turbidimetrically; TC, TG, high-density lipoprotein (HDL), and GGT were determined via enzymatic, colorimetric method. All tests were performed on a Roche/Hitachi cobas c701/c502 analyzers (Roche Diagnostics, Indianapolis, IN, USA). LDL and VLDL were calculated using a new equation developed by the National Heart, Lung and Blood Institutes of Health which overcomes the limitations of the existing Friedewald LDL equation and demonstrates validity in fasting individuals [14].

### 2.7. SOS-Free Diet Screener

The SOS-Free Diet Screener is a 27-question dietary assessment tool that we developed to measure adherence to an exclusively whole-plant-food diet free of added salt, oil, and sugar (SOS-Free Diet, see screener in Appendix A). The screener asks participants to indicate how often (0 per month, 1 per month, 2–3 per month, 1 per week, 2 per week, 3–4 per week, 5–6 per week, 1 per day, 2 per day, 3 per day, 4 per day, ≥5 per day) they consumed a serving of the specified food group, ingredient, beverage, or dietary supplement over the previous 30 days. Questions include multiple categories of minimally processed plant and animal foods, highly processed foods, foods containing added salt, oil, and sugar, alcoholic and caffeinated beverages, dietary supplements, and tobacco. Serving recommendations of plant foods used for scoring (see scoring key in Appendix A) meet or exceed recommendations of dietary indices (AHEI, aMED, DASH) used in current nutrition research that measure the potential of a diet to reduce the risks of chronic disease and all-cause mortality [15].

The scoring key is designed to capture select foods in the diet and follows a proposed standardized methodology for measuring dietary adherence [16]. Screener data were used to calculate a non-adherence score where zero points represents 100% adherence to the SOS-Free Diet and each point above zero indicates one serving out-of-compliance, either one too few servings of a recommended food or one too many servings of a discouraged food to a maximum of 82.17 points, which would indicate 0% adherence. The non-adherence score was calculated by first converting the raw intake data into daily intake frequencies and then using the equation detailed in Appendix A.

### 2.8. Statistical Analysis

Descriptive statistics were reported for all clinical parameters and continuous metrics were summarized by median and interquartile range (IQR). For all clinical parameters, regression coefficients were presented with 95% credible intervals (CI). In this report, a regression coefficient is considered significant when the CI does not include zero. The Bayesian framework was used for the statistical analysis due to the small sample size and the availability of baseline, EOF, and EOR data from a different population in a previous study [6]. This prior information was incorporated into the model by incorporating the current data and the prior data to the model all at once [17] using Bayesian Regression Models in Stan (brms) default priors and extending the model to include an indicator variable to distinguish current and prior data. With smaller sample sizes and longitudinal designs, Bayesian estimation with informative priors performs better than maximum-likelihood estimation and restricted-maximum-likelihood-estimation [18].

The primary goal for each clinical parameter was to estimate the difference between values at each time point (i.e., baseline, EOF, EOR, and FU) and determine which differences were significant. This was investigated using a random intercept mixed effect model with the clinical parameter as the dependent variable and participant ID as the grouping variable. The main fixed effect of interest was the study visit time point (i.e., baseline, EOF, EOR, and FU). Age, sex, and a prior data indicator were used in the models as fixed effect control variables. For each parameter in the model, the Markov Chain Monte Carlo (MCMC) convergence was assessed using Rhat with a threshold of 1.01 [19]. Moreover, traceplots were examined for evidence that Markov chains were well mixed [20]. Posterior predictive checks were used to test whether data simulated from the fitted model were similar to the observed data [21]. Random intercept models that did not perform well on diagnostics were re-run after natural log transforming the dependent variable. For models that used informative priors, sensitivity analyses were examined by running the models without including prior data and without including the prior fixed effect.

## 3. Results

We enrolled 40 participants, of which 38 completed the EOF and EOR visits and 33 completed the FU visit (see Methods and Figure 1). Of the 38 participants who completed the water-only fasting and refeeding protocol, 31 were female and 7 were male. The median (IQR) age was 60 (52, 65) years. The median (IQR) water-only fasting and refeeding lengths were 14 (13, 19) and 6 (4, 8), respectively. The FU visit occurred at a median (IQR) of 45 (43, 50) days after completion of treatment (Table 1).

During the period between the EOF and FU visits, participants were encouraged to eat an ad libitum SOS-Free Diet. Adherence was assessed using a dietary screener specific for this diet. A non-adherence score was calculated such that 100% adherence is equal to a score of zero and 0% adherence is equal to a score of 82 (see Methods). At baseline, the median (IQR) non-adherence score was 10 (7, 16) points. At FU, the non-adherence score had decreased slightly to 6 (3, 8) points (Table 2). Participants reported an increase in daily servings of vegetables, fruits, and legumes and a reduction in daily servings of animal protein, dairy, eggs, refined grain flour, salt, oil, and sugar. (Appendix A).

There were clinically meaningful reductions in median BW, BMI, and AC by the EOF visit which were sustained at the EOR and FU visits (Table 3 and Figure 2). Differences in BW, BMI, and AC were significant at the EOF, EOR, and FU visits compared to the baseline visit. At the FU visit, the estimated differences (95% CI) in BW, BMI, and AC were −7.52 kg (−8.43 kg, −6.61 kg), −2.70 kg/m^2^ (−3.01 kg/m^2^, −2.39 kg/m^2^), and −6.58 cm (−7.68 cm, −5.48 cm), respectively (Table 4).

There were also clinically meaningful reductions in median SBP and DBP that were sustained at the FU visit (Table 3 and Appendix A). Differences in SBP were significant at the EOF, EOR, and FU visits compared to the baseline visit, and differences in DBP were significant at the EOR and FU visits compared to the baseline visit. At the FU visit, the estimated differences (95% CI) in SBP and DBP were −7.68 mmHg (−12.31 mmHg, −3.04 mmHg) and −2.44 mmHg (−4.92 mmHg, −0.01 mmHg), respectively (Table 4).

There were no differences in median TC nor LDL between the EOF and baseline visits but there was a decrease at the EOR and FU visits compared to the baseline visit (Table 3 and Appendix A). At the FU visit, the estimated differences (95% CI) in TC and LDL were −0.35 mmol/L (−0.63 mmol/L, −0.07 mmol/L) and −0.34 mmol/L (−0.59 mmol/L, −0.09 mmol/L), respectively (Table 4). There was a decrease in median HDL at the EOF and EOR visits compared to the baseline visit, which increased between the FU and EOR visits (Table 3). Median HDL values remained within normal clinical reference interval (>1.01 mmol/L) for the duration of the study (Table 3). There were significant increases in VLDL and TG between the EOR and baseline visits and reductions between the FU and EOR visits, but there were no differences between the FU and baseline visits (Table 3 and Table 4).

There was a decrease in median glucose, insulin, and HOMA-IR at the EOF visit and a subsequent increase at the EOR visit (Table 3, Figure 3 and Appendix A). The estimated difference (95% CI) in log transformed HOMA-IR values between the EOF and baseline visits was −0.58 (−0.76, −0.41), which increased after refeeding by 0.54 (0.36, 0.71) above the baseline visit and by 1.12 (0.94, 1.30) above the EOF visit (Table 4). A principal aim of this study was to determine if the previously reported increase in HOMA-IR after refeeding was a sustained or temporary phenomenon [6]. At the FU visit, median glucose, insulin, and HOMA-IR had significantly decreased from the EOR visit and slightly decreased from the baseline visit (Table 3). The estimated difference (95% CI) in log transformed HOMA-IR between the FU and EOR visits was −0.63 (−0.85, −0.40) (Table 4).

We also assessed inflammation by measuring hsCRP and calculating FLI with an equation using BMI, AC, TG, and GGT. Median hsCRP increased between the EOF and baseline visits and decreased at the EOR and FU visits compared to the baseline visit (Table 3). At the FU visit, the estimated difference (95% CI) in log transformed hsCRP was −0.42 mg/L (−0.67 mg/L, −0.17 mg/L) (Table 4). There were significant reductions in FLI at the EOF, EOR, and FU visits compared to the baseline visit (Table 3 and Figure 2). At the FU visit, the estimated difference (95% CI) from baseline was −11.80 (−15.49, −8.11) (Table 4).

## 4. Discussion

We previously found that prolonged water-only fasting followed by an exclusively whole-plant-food refeeding diet improved several biomarkers correlated with increased risk of cardiometabolic disease but also increased insulin resistance in non-diabetic, overweight/obese participants [6]. The results presented here confirm our previous findings and expand the analysis by assessing the effects six weeks after the completion of refeeding. Our data suggest that water-only fasting followed by ad libitum consumption of an exclusively whole-plant-food diet results in sustained biomarker improvement and that the insulin resistance observed during refeeding appears to be transient. We speculate that the temporary insulin resistance observed during refeeding is consistent with prior reports of fasting-induced insulin resistance during the metabolic switch from ketones to glucose [23].

We found that approximately two weeks of combined water-only fasting and refeeding resulted in an estimated BW reduction of 7.5 kg. Accordingly, median baseline BMI dropped from the obese to the overweight category. Additionally, we have forthcoming data from the same population that the 6% decrease in AC may be associated with a loss of visceral adipose tissue, which is associated with a reduction in whole-body insulin resistance. Methods of fasting or calorie restriction, including water-only fasting, have been criticized for producing unsustainable changes in BW [4]. However, we found that reductions in BW, BMI, and AC were all sustained for at least six weeks following the fasting and refeeding period (Figure 2). Long-term follow-up data is lacking, but there does not appear to be a substantial difference in weight loss between caloric restriction, time restricted eating, and intermittent fasting interventions lasting 8 or more weeks [4,11,24]. A recent umbrella review of 11 meta-analyses reported a mean difference in weight of less than 2 kg for intermittent fasting interventions, with the highest level of evidence for modified alternate-day fasting [25]. There have been no studies directly comparing water-only fasting to caloric restriction, time restricted eating, intermittent fasting, or other prolonged fasting interventions, but our results suggest that water-only fasting followed by an ad libitum whole-plant-food diet may be an effective way to achieve sustained weight loss as a means for disease prevention and reversal.

Our results support our previous findings that median TC, LDL, VLDL, and TG did not change immediately after fasting, but there was a decrease in median TC and LDL and an increase in median VLDL and TG after refeeding. Additionally, we observed that the decreases in TC and LDL were sustained while the increase in VLDL and TG decreased to within normal clinical range at the six-week FU (Appendix A). The metabolic switch to ketogenesis during fasting increases lipolysis, de novo synthesis of TG in the liver, and localized adipose tissue inflammation, all of which may lead to an increase in circulating lipids [26]. Recent publications assessing the effects of similar fasting protocols on lipids have reported inconsistent lipid mobilization trends after fasting and refeeding [9,10,27], which may highlight the heterogeneity of glucose and lipid metabolism in obese populations [28]. Furthermore, it is theorized that during periods of fasting and increased physical activity, muscle tissue utilizes TG rather than glucose as an energy source, and therefore increased TG during fasting may have a beneficial rather than harmful effect as it does during periods of non-fasting or chronic inactivity [29]. Although the mechanism of lipid utilization is beyond the scope of this study, overall lipid mobilization during fasting may improve dyslipidemia, decrease diabetes risk [30], and reduce protein catabolism during prolonged fasting.

HOMA-IR is calculated using fasting glucose and insulin and used to approximate insulin sensitivity, with values above 2.0 indicating insulin resistance [12]. As previously reported, we observed that median HOMA-IR increased above 2.0 after a minimum of five days on the post-fast refeeding diet, suggesting that participants developed insulin resistance. To determine if this increase was sustained or transient, we assessed fasting glucose and insulin six weeks after the EOR visit and found that median HOMA-IR had decreased to slightly below the median baseline value (Appendix A). The return of HOMA-IR to baseline levels indicates that the insulin resistance observed after refeeding is transient and likely adaptive rather than pathologic. Similar to our findings, a recent study on normal weight adults undergoing five days of water-only fasting followed by three days of refeeding with a rice-based diet reported increased insulin and HOMA-IR at the end of the refeeding period [10]. Research comparing different refeeding diet protocols is lacking and the effect of refeeding diet on post-fast insulin response is unknown.

Fasting causes a well-defined transition from glucose metabolism to fatty acid oxidation [11,31] with the majority of cells efficiently metabolizing ketones as an alternative energy source. During prolonged fasting, rates of hepatic and renal gluconeogenesis also increase in order to meet any remaining glucose needs. There is a lack of research into the transition from fatty acid oxidation back to glucose metabolism, but it is known that insulin reduces hepatic gluconeogenesis [32]. It may be that increased insulin during the post-fast refeeding period is an adaptive response to decrease gluconeogenesis or that it facilitates glucose uptake in tissue, such as muscle, that have adapted to fatty acid metabolism [29]. Further research into the immediate post-fast refeeding period is necessary to answer outstanding questions regarding the adaptive changes that occur and how refeeding diet affects these changes.

FLI is another strong predictor of metabolic disease and indicates an increased risk of developing T2D independent of insulin resistance [13]. The use of this index may also provide insight into fat metabolism and liver health during fasting and refeeding. Our results show that compared to baseline, there was a reduction in median FLI at every time point (EOF, EOR, and FU). As mentioned above, metabolic adaptations during fasting occur in multiple organ systems, such as the liver, adipose tissue, and skeletal muscle. It is important to understand how these organ systems interact, as there is a strong interplay between blood glucose, fatty acid homeostasis, and inflammation, both localized and systemic, each of which contributes to metabolic disease. Fasting and other forms of caloric restriction have been shown to correspond with a mild increase in inflammatory markers and may increase adipose tissue inflammation by increasing macrophage infiltration [27,33]. Although we found that median hsCRP remained within acceptable clinical range, there was a mild increase at EOF while median hsCRP was lower than baseline at EOR and FU. These effects suggest that fasting may have a hormetic and potentially anti-inflammatory effect. The precise action of fasting on inflammatory response should be further explored.

The participants in this study were educated and instructed to follow an SOS-Free Diet during the refeeding and FU period. Dietary adherence was assessed at baseline and FU using a dietary screener that was specifically designed to measure adherence to the recommended diet but has not yet been validated. This diet is purported to promote health and reverse disease, and may potentiate or prolong the results achieved by fasting alone. Diets predominant in plant foods decrease obesity-related inflammatory markers, such as CRP and IL-6, which are associated with disease [34]. Furthermore, vegetables, fruits, whole grains, and legumes may prevent chronic disease [35]. A main criticism of restrictive diets is their lack of long-term adherence [4], but we found that six weeks after returning home, there was an improvement in dietary adherence. This may be due to the potential effects of fasting on taste adaptation, particularly on salty and sweet food sensitivity, which may reduce the need for added salt, oil, or sugar, and allow for better adherence [36]. There are still unanswered questions about how an SOS-Free Diet impacts short- and long-term health outcomes, and whether individual dietary changes such as the addition of leafy green vegetables versus the elimination of added salt, oil, or sugar, impact health.

It is widely reported that humans have practiced therapeutic water-only fasting for more than two thousand years and there is more than a century of published literature on the physiological and clinical effects of water-only fasting in humans. Over the past decade, clinical research into the beneficial and adverse health effects of water-only fasting has progressed with a level of scientific rigor not previously reported [6,7,8,9,10,26,36]. Nonetheless, there is still a concern about the safety and practicality of prolonged water-only fasting in humans [4]. In this study, 95% (38/40) of the enrolled participants were able to complete at least 10 days of fasting with only mild (grade 1) to moderate (grade 2) adverse events. This suggests that—at least in a well-selected population—fasting is well-tolerated. Furthermore, modifications to water-only fasting, such as the inclusion of vegetable broth and/or juice or the temporary suspension of fasting with easily digestible foods, are commonly implemented as part of clinical practice to ensure patient safety, comfort, and well-being and do not appear to have a negative impact on treatment outcomes [6,8].

This open-label, observational study has several limitations including that it occurred at a single fasting center and had a small sample size. Due to the small sample size, the group was not differentiated into any demographic categories such as age, gender, or ethnicity. Another limitation is that the study lacked normal-weight, metabolically “healthy” and diabetic control groups, which are necessary to better understand the effects of body weight and dysregulated glucose metabolism on insulin sensitivity following prolonged fasting. Furthermore, the study lacked a dietary intervention group, which is necessary to assess the effect of the SOS-Free Diet on cardiometabolic health markers and to determine if the sustained effects are attributable to diet and/or the hormetic effects of water-only fasting. An additional limitation is that the SOS-Free Dietary screener has not been validated. Further development and validation of this screener is currently underway.

## 5. Conclusions

Metabolic diseases, particularly CVD and T2D, are a global public health concern and there is increased interest in developing and implementing cost-effective methods of prevention and reversal. The pathophysiology of metabolic disease is complex, and there is variability in how obesity impacts insulin resistance and lipid metabolism, as some obese people remain insulin sensitive without associated cardiometabolic disorders [26,28]. Nevertheless, early intervention with treatments that result in sustained improvements in biomarkers, such as BW, that correlate with increased cardiometabolic disease risk may prevent the development of these chronic conditions and improve overall health. Water-only fasting improves obesity as well as other markers of CVD risk including lipid profile, FLI, and hsCRP, and the results are sustained for at least six weeks with imperfect adherence to an exclusively whole-plant-food diet. This study is encouraging and sets a precedent for future research into this intervention as a potential treatment for efficient and sustained weight loss and improvement in markers of CVD risk.

## Figures and Tables

**Figure 1 nutrients-14-04313-f001:**
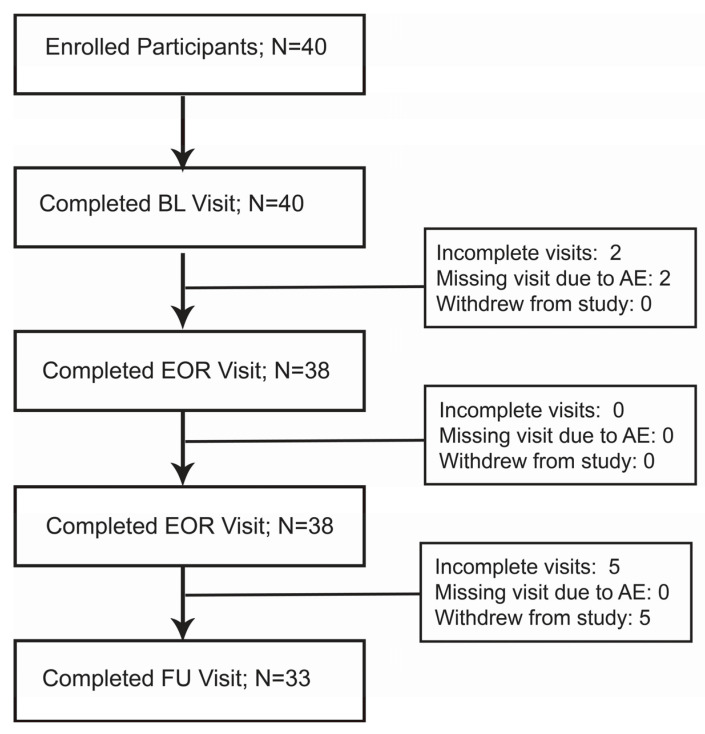
Enrollment and participation flow diagram. EOF, End of fast; EOR, end of refeed; FU, follow up; AE, adverse event. See results section for details of specific AEs.

**Figure 2 nutrients-14-04313-f002:**
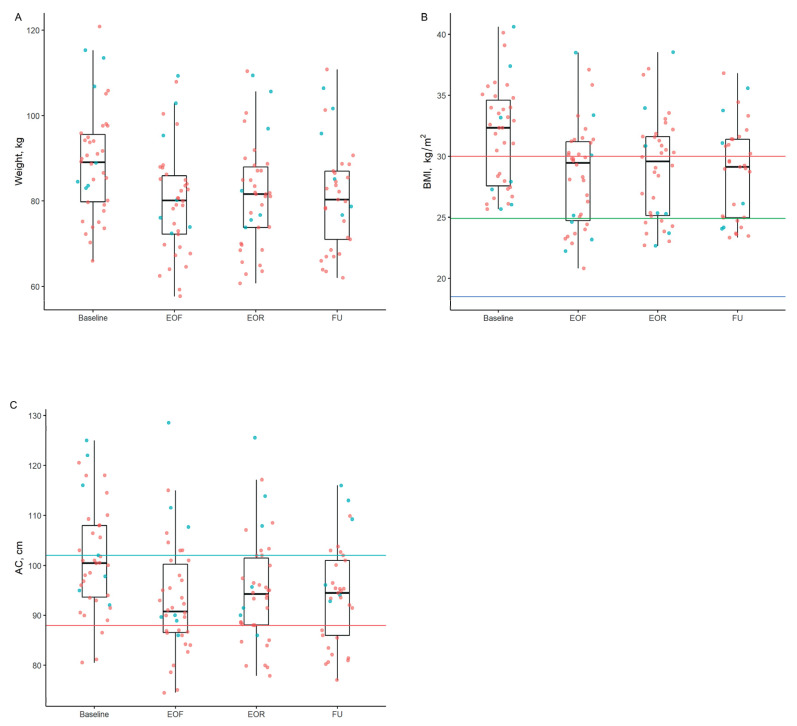
Boxplots of Body Weight, BMI, and AC at Baseline, EOF, EOR, and FU. Box-plot distribution of (**A**) body weight (kg), (**B**) BMI (kg/m^2^), and (**C**) AC. BMI above 30 kg/m^2^ (red line) is obese, BMI between 25 (green line) and 29 kg/m^2^ is overweight, and BMI between 18 and 24 kg/m^2^ is normal. AC above102 cm (blue) denotes upper threshold for males and above 88 cm (pink line) denotes upper threshold for females [22]. Boxplots include the minimum value, first (**lower**) and third (**upper**) quartiles, the median, and the maximum value. On all plots, blue dots represent male participants and pink dots represent female participants. EOF, end of fast; EOR, end of refeed, FU, follow up; BMI, body mass index; AC, abdominal circumference; kg, kilogram; m, meter; cm, centimeter.

**Figure 3 nutrients-14-04313-f003:**
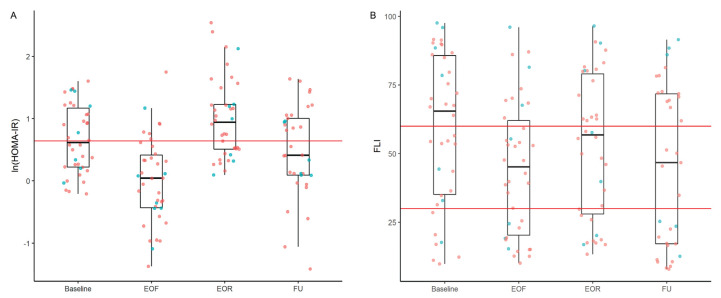
Boxplots of Metabolic Health Indicis at Baseline, EOF, EOR, and FU. Box-plot distribution of (**A**) ln(HOMA-IR) where values below 0.64 (red line) is optimal insulin sensitivity [12]. ln indicates that HOMA-IR values and reference values were natural log transformed and (**B**) FLI where values between 0–30 (red line) are desirable and values equal to or above 60 (red line) denote presence of fatty liver disease [13]. Boxplots include the minimum value, first (**lower**) and third (**upper**) quartiles, the median, and the maximum value. On all plots, blue dots represent male participants, and pink dots represent female participants. EOF, end of fast; EOR, end of refeed; FU, follow up; HOMA-IR, homeostatic model assessment for insulin resistance; FLI, fatty liver index.

**Table 1 nutrients-14-04313-t001:** Participant and Visit Characteristics.

Characteristic	Overall (*N* = 38)	Female (*n* = 31)	Male (*n* = 7)
Age, y	60 (52, 65)	61 (53, 66)	59 (51, 62)
Fast Length, d	14 (13, 19)	14 (13, 19)	14 (12, 15)
Refeed Length, d	6 (4, 8)	6 (5, 8)	5 (4, 6)
Follow-up Length^λ^, d	45 (43, 50)	46 (44, 50)	43 (41, 47)

Data presented as Median (Interquartile Range). y, years; d, day; ^λ^ Follow-up length had 5 overall missing values, 4 missing values for females, and 1 missing value for males.

**Table 2 nutrients-14-04313-t002:** Dietary Screener Non-Adherence Score.

	BL	FU
Total Non-Adherence Score	10 (7, 16)	6 (3, 8)

Data is presented as median (interquartile range). BL, baseline FU, follow-up.

**Table 3 nutrients-14-04313-t003:** Effect of Fasting, Refeeding, and Follow-up on Cardiometabolic Markers.

	Median (IQR)			
	BL	EOF	EOR	FU
BW, kg	89.1 (79.8, 95.6)	80.1 (72.2, 85.9)	81.6 (73.8, 88.0)	80.3 (71.0, 87.0)
BMI, kg/m^2^ (18.5–24.9 kg/m^2^)	32.3 (27.6, 34.6)	29.5 (24.7, 31.2)	29.6 (25.1, 31.6)	29.1 (25.0, 31.4)
AC, cm (<88 cm for women)	100.4 (93.2, 107.2)	91.0 (86.2, 97.5)	94.0 (88.0, 98.7)	93.6 (84.5, 98.3)
AC, cm (<102 cm for men)	102.0 (96.4, 119.0)	90.0 (89.3, 109.6)	95.7 (90.8, 110.8)	102.7 (94.5, 112.0)
SBP, mmHg (<120 mmHg)	123 (111, 131)	113 (105, 125)	110 (103, 115)	114 (110, 122)
DBP, mmHg (<80 mmHg)	78 (73, 85)	80 (73, 85)	76 (70, 79)	78 (71, 80)
TC, mmol/L (2.59–5.15 mmol/L)	5.10 (4.38, 5.56)	5.06 (4.36, 5.55)	4.65 (4.01, 5.30)	4.61 (4.27, 5.36)
LDL, mmol/L (<2.56 mmol/L)	3.17 (2.51, 3.76)	3.28 (2.65, 3.89)	2.72 (2.02, 3.21)	2.72 (2.43, 3.37)
HDL, mmol/L (>1.01 mmol/L)	1.29 (1.04, 1.62)	1.15 (0.96, 1.30)	1.14 (0.96, 1.33)	1.19 (0.96, 1.48)
VLDL, mmol/L (0.13–1.04 mmol/L)	0.49 (0.44, 0.64)	0.60 (0.54, 0.69)	0.69 (0.58, 0.92)	0.57 (0.41, 0.78)
TG, mmol/L (< 3.86 mmol/L)	1.16 (1.05, 1.56)	1.48 (1.29, 1.65)	1.68 (1.40, 2.32)	1.45 (1.01, 1.88)
Glucose, mmol/L (3.61–5.49 mmol/L)	5.19 (4.79, 5.59)	4.38 (3.89, 4.61)	5.47 (5.12, 5.83)	5.00 (4.72, 5.38)
Insulin, pmol/L (15.6–149.4 pmol/L)	50 (33, 76)	31 (24, 43)	62 (41, 90)	41 (31, 72)
Insulin ^‡^, pmol/L (2.75–5.01 pmol/L)	3.92 (3.51, 4.34)	3.43 (3.16, 3.76)	4.12 (3.72, 4.50)	3.72 (3.42, 4.28)
HOMA-IR (<1.9 insulin sensitive)	1.85 (1.25, 3.22)	1.04 (0.65, 1.51)	2.56 (1.66, 3.41)	1.51 (1.10, 2.73)
HOMA-IR ^‡^ (<0.64 insulin sensitive)	0.61 (0.22, 1.17)	0.04 (−0.43, 0.41)	0.94 (0.51, 1.23)	0.41 (0.09, 1.00)
GGT ^ζ,^ nmol/(s*L) (<1000 nmol/(s*L))	250 (200, 333)	250 (200, 283)	250 (200, 367)	233 (167, 317)
GGT ^‡ζ,^ nmol/(s*L) (<6.91 nmol/(s*L)	5.52 (5.30, 5.81)	5.52 (5.30, 5.65)	5.52 (5.30, 5.90)	5.45 (5.12, 5.76)
FLI ^ζ^ (<30 is optimal)	65 (35, 86)	45 (20, 62)	57 (28, 79)	47 (17, 72)
hsCRP, mg/L (<3 mg/L)	1.83 (0.98, 3.81)	2.63 (1.46, 5.10)	1.06 (0.54, 2.00)	0.90 (0.45, 2.14)
hsCRP^‡^, mg/L (<1.10 mg/L)	0.61 (−0.03, 1.34)	0.97 (0.38, 1.63)	0.05 (−0.62, 0.69)	−0.11 (−0.80, 0.76)

Labcorp reference ranges for normal values are provided below the respective variable. N = 38 at BL, EOF, and EOR. n = 33 at FU. IQR, interquartile range; BL, baseline; EOF, end of fast; EOR, end of refeed; FU, follow up; BW, body weight; AC, abdominal circumference; SBP, systolic blood pressure; DBP, diastolic blood pressure; TC, total cholesterol; LDL, low-density lipoprotein; HDL, high-density lipoprotein; VLDL, very-low density lipoprotein; TG, triglycerides; HOMA-IR, homeostatic model assessment for insulin resistance; GGT, gamma-glutamyl transferase; FLI, fatty liver index; hsCRP, high-sensitivity C-reactive protein; kg, kilogram; m, meter; cm, centimeter; mmHg, millimeters of mercury; mmol/L, millimole per liter; pmol/L, picomole per liter; nmol/(s*L), nanomole per second liter; mg/L, milligram per liter. ^‡^ Experimental and reference values are natural log transformed. ^ζ^ Analysis used default priors.

**Table 4 nutrients-14-04313-t004:** Significance of Differences for Changes in Cardiometabolic Markers.

	EOF − BL	EOR − BL	FU − BL	EOR − EOF	FU − EOF	FU − EOR
	Estimate (95% CI)					
BW, kg	−8.98 * (−9.69, −8.25)	−7.48 * (−8.19, −6.78)	−7.52 * (−8.43, −6.61)	1.50 * (0.79 2.22)	1.46 * (0.55, 2.37)	−0.04 (−0.93, 0.87)
BMI, kg/m^2^	−3.20 * (−3.45, −2.96)	−2.68 * (−2.92, −2.43)	−2.70 * (−3.01, −2.39)	0.53 * (0.27, 0.78)	0.50 * (0.19, 0.82)	−0.02 (−0.34, 0.30)
AC, cm	−8.13 * (−9.00, −7.26)	−6.51 * (−7.36, −5.67)	−6.58 * (−7.68, −5.48)	1.62 * (0.78, 2.48)	1.55 * (0.44, 2.66)	−0.07 (−1.16, 1.03)
SBP, mmHg	−11.49 * (−15.17, −7.84)	−14.58 * (−18.28, −10.91)	−7.68 * (−12.31, −3.04)	−3.09 (−6.70, 0.55)	3.81 (−0.82, 8.40)	6.90 * (2.19, 11.54)
DBP, mmHg	−0.54 (−2.40, 1.32)	−3.33 * (−5.18, −1.48)	−2.44 * (−4.92, −0.01)	−2.79 * (−4.67, −0.89)	−1.90 (−4.38, 0.60)	0.88 (−1.56, 3.31)
TC, mmol/L	−0.04 (−0.26, 0.18)	−0.51 * (−0.72, −0.29)	−0.35 * (−0.63, −0.07)	−0.47 * (−0.68, −0.25)	−0.31 * (−0.59, −0.03)	0.16 (−0.12, 0.44)
HDL, mmol/L	−0.16 * (−0.22, −0.10)	−0.15 * (−0.21, −0.09)	−0.05 (−0.12, 0.03)	0.01 (−0.05, 0.07)	0.11 * (0.03, 0.19)	0.10 * (0.02, 0.18)
LDL, mmol/L	0.09 (−0.11, 0.29)	−0.59 * (−0.79, −0.39)	−0.34 * (−0.59, −0.09)	−0.68 * (−0.88, −0.48)	−0.42 * (−0.67, −0.18)	0.26 * (0.00, 0.51)
VLDL, mmol/L	0.03 (−0.03, 0.09)	0.23 * (0.17, 0.29)	0.04 (−0.04, 0.11)	0.20 * (0.14, 0.26)	0.01 (−0.07, 0.08)	−0.20 * (−0.27, −0.12)
TG, mmol/L	0.08 (−0.07, 0.22)	0.58 * (0.43 0.73)	0.08 (−0.11, 0.27)	0.50 * (0.36, 0.65)	−0.00 (−0.18, 0.19)	−0.50 * (−0.69, −0.31)
Glucose, mmol/L	−0.79 * (−0.97, −0.61)	0.53 * (0.35, 0.71)	−0.13 (−0.36, 0.10)	1.32 * (1.13, 1.50)	0.66 * (0.43, 0.89)	−0.66 * (−0.89, −0.43)
Insulin ^‡^, pmol/L	−0.42 * (−0.58, −0.26)	0.44 * (0.29, 0.60)	−0.06 (−0.27, 0.14)	0.86 * (0.70, 1.02)	0.35 * (0.15, 0.55)	−0.51 * (−0.71, −0.31)
HOMA-IR ^‡^	−0.58 * (−0.76, −0.41)	0.54 * (0.36, 0.71)	−0.09 (−0.32, 0.14)	1.12 * (0.94, 1.30)	0.50 * (0.27, 0.72)	−0.63 * (−0.85, −0.40)
GGT ^‡ζ^, nmol/(s*L)	−0.06 (−0.16, 0.05)	0.02 (−0.08, 0.12)	−0.09 (−0.19, 0.02)	0.07 (−0.03, 0.18)	−0.03 (−0.14, 0.08)	−0.10 (−0.21, 0.00)
FLI ^ζ^	−14.62 * (−18.09, −11.07)	−6.12 * (−9.67, −2.59)	−11.80 * (−15.49, −8.11)	8.49 * (4.96, 12.01)	2.82 (−0.81, 6.52)	−5.67 * (−9.38, −1.93)
hsCRP ^‡^, mg/L	0.29 * (0.10, 0.49)	−0.48 * (−0.68, −0.29)	−0.42 * (−0.67, −0.17)	−0.77 * (−0.97, −0.57)	−0.71 * (−0.97, −0.45)	0.06 (−0.19, 0.31)

Estimated significance of difference and 95% credible intervals established using a Bayesian framework. CI, credible intervals; BL, baseline; EOF, end of fast; EOR, end of refeed; FU, follow up; BW, body weight; AC, abdominal circumference; SBP, systolic blood pressure; DBP, diastolic blood pressure; TC, total cholesterol; LDL, low-density lipoprotein; HDL, high-density lipoprotein; VLDL, very-low density lipoprotein; TG, triglycerides; HOMA-IR, homeostatic model assessment for insulin resistance; GGT, gamma-glutamyl transferase; FLI, fatty liver index; hsCRP, high-sensitivity C-reactive protein; kg, kilogram; m, meter; cm, centimeter; mmHg, millimeters of mercury; mmol/L, millimole per liter; pmol/L, picomole per liter; nmol/(s*L), nanomole per second liter; mg/L, milligram per liter. * Zero lies outside the 95% CI so the finding is considered significant. ^‡^ Experimental and reference values are natural log transformed. ^ζ^ Analysis used default priors.

## Data Availability

Data is available upon request to the corresponding author.

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
