# Peer review of "A Six-Week Follow-Up Study on the Sustained Effects of Prolonged Water-Only Fasting and Refeeding on Markers of Cardiometabolic Risk"

_nutrients, 2022, doi:10.3390/nu14204313_

Round 1

Reviewer 1 Report

1.The study design is dangerous and not suitable for the human. Although only 3-4 participants suffered from nausea, the electrolyte status and blood sugar did not be monitor closely during 14 days fasting.

2. Those parameters are fluctuated by times, it might be over-explain the long term effect.

3.Fasting might reduce the insulin resistance by this study, however, we cannot see the long term effect by starvation of this 6 weeks study.

4. The study design should be follow up longer to get the conclusion.

Author Response

1.The study design is dangerous and not suitable for the human. Although only 3-4 participants suffered from nausea, the electrolyte status and blood sugar did not be monitor closely during 14 days fasting.

Your concerns regarding the safety of water-only fasting in humans are greatly appreciated. In response, we will discuss key points that support the general safety of this study as well as the use of water-only fasting as a human therapeutic intervention.

The safety of the research participants volunteering in this study was our primary concern, and we took necessary steps to assure their safety. As indicated in our manuscript’s Institutional Review Board Statement (see below), and in compliance with clinical research regulations, the protocol was reviewed, approved, and monitored by a federally recognized institutional review board.

Institutional Review Board Statement: The study was conducted according to the guidelines of the Declaration of Helsinki, and approved by the Institutional Review Board of the TrueNorth Health Foundation (TNHF-2020-2VAT, April 2, 2020).

Additionally, the study was conducted as an observational study and as such, the water-only fasting intervention was implemented at a residential, medical fasting center that employs trained medical personnel and offers 24-hour clinical care. The center has been in operation for over thirty years and has supervised over 20,000 water-only fasting patients without any deaths or long-term negative health effects. The center follows a specific protocol that includes comprehensively screening patients for contraindications before they are approved to water-only fast. If patients are approved to water-only fast, their vital signs are monitored twice daily and they receive weekly urinalysis and serology monitoring that includes comprehensive metabolic profile (electrolytes and glucose) and complete blood count as well as any other required patient-specific labs. If a patient has any abnormal electrolyte or glucose values then the fast may be temporarily interrupted with therapeutic broth or terminated with the approved refeeding protocol. The water-only fasting protocol implemented at the center was published in a retrospective study which assessed safety and reported on adverse events occurring in greater than 700 patients who water-only fasted between 2 and 40 days [8]. The study found that the vast majority of adverse events (95%) were mild to moderate in nature and that the risk of having a serious adverse event was less than .05%. Furthermore, there have been several studies published recently that implement water-only fasting periods of similar length [6-10, 26, 27, 36].

The participants in this study were recruited from patients of the aforementioned fasting center who were already planning on water-only fasting the minimum fast length before they were contacted by research personnel. The patients were also approved to water-only fast the minimum fast length by non-research medical personnel before they were enrolled in the study. The research protocol did not interfere with the water-only fasting intervention (i.e., length, monitoring, etc.) in any way.

We have added the following to the methods section clarify the safety measures of the fasting:

Ethical Statement

This study (NCT04514146) was conducted according to the guidelines of the Declaration of Helsinki, and approved by the Institutional Review Board of the TrueNorth Health Foundation (TNHF-2020-2VAT).

Medically Supervised, Water-only Fasting & Refeeding Protocol

Participants were instructed to consume a minimum of 40 ounces of distilled water per day and limit physical activity during the course of the fast. While fasting, participants remained on-site and vital signs and symptoms were monitored twice daily by non-research, medical personnel, along with weekly serology and urine analysis to monitor electrolyte balance and other physiological functions, such as kidney and liver function. Adverse events were continuously monitored and if necessary the fast was temporarily interrupted with vegetable broth or juice or suspended by initiating the standard refeeding protocol

Additionally this is clarified in the discussion:

In this study, 95% (38/40) of the enrolled participants were able to complete at least 10 days of fasting with only mild (grade 1) to moderate (grade 2) adverse events. This suggests that - at least in a well-selected population - fasting is well-tolerated. Furthermore, modifications to water-only fasting, such as the inclusion of vegetable broth and/or juice or the temporary suspension of fasting with easily digestible foods, are commonly implemented as part of clinical practice to ensure patient safety, comfort, and well-being and do not appear to have a negative impact on treatment outcomes [6, 8].

  1. Those parameters are fluctuated by times, it might be over-explain the long term effect.

Thank you for identifying the discrepancy in using the term “long-term”, we agree and have made changes to the title and throughout the text to clarify that follow-up occurred at “six-weeks” and that further investigation may be necessary to determine “long-term” effects. To our knowledge, this study is novel in that it provides at least six-weeks post-fast data on select biomarkers. 

New title reads as follows:

A Six-Week Follow-up Study on the Sustained Effects of Prolonged Water-only Fasting and Refeeding on Markers of Cardiometabolic Risk

3.Fasting might reduce the insulin resistance by this study, however, we cannot see the long term effect by starvation of this 6 weeks study.

To be clear, the participants in this study never entered starvation. Starvation, is defined as a chronic state of nutritional deficiency during which the body uses essential tissues for fuel, especially protein, and as such should be distinguished from the total fasted state that was observed in this study [8].

This study did not attempt to report on long-term clinical health outcomes, but in an attempt to better understand acute physiological changes as well as sustained effects on select biomarkers. We reported on select variables after fasting, refeeding, and six-weeks post-fast. We chose a follow-up period of six weeks because based on clinical observation this should be a sufficient amount of time for any acute changes that are solely the result of physiological adaptation to the total fasted state to have stabilized thus allowing us to observe sustained effects on select biomarkers.

Of course, the data does not indicate if these changes will last beyond six weeks nor do we know if the changes are caused or supported by other lifestyle factors (i.e., dietary change) as we were unable to include a dietary control group, which we clearly describe as a study limitation. Nevertheless, we found the sustained improvement in markers of cardiometabolic health for at least six weeks to be promising results.

As stated in the previous question, we have made changes to the title and throughout the text to reflect that follow-up occurred at “six-weeks”.

  1. The study design should be follow up longer to get the conclusion.

It should be noted that this was a preliminary, follow-up, observational study to a previously published study that described changes only until the refeeding phase [6]. The goal was to describe changes in markers of cardiometabolic risk six-weeks after the completion of the fast. This study was not interventional nor did it have appropriate randomized controls to draw conclusions on short- or long-term health outcomes.

Furthermore, it is particularly difficult to conduct this type of study because the cardiometabolic biomarkers we reported on are also affected by lifestyle variables such as diet. Therefore, we chose a follow-up time point of six weeks because it was determined to be a sufficient amount of time for post-fast stabilization to occur but not so long that any potential benefits or harms resulting from water-only fasting would be missed. As previously mentioned and clearly discussed as a limitation.

In order to clarify this distinction, we have made changes to the title and throughout the text to clarify that follow-up occurred at “six-weeks”.

Reviewer 2 Report

Dear authors,

could you please describe scientific soundness of water-only fasting in more detailed way in Discussions?

Was the study approved by Ethical committee or board? Was written consent obtained from the patients to participate in the research study?

Author Response

  1. Could you please describe scientific soundness of water-only fasting in more detailed way in Discussions?

We have made additions to the discussion as follows:

It is widely reported that humans have practiced therapeutic water-only fasting for more than two thousand years and there is more than a century of published literature on the physiological and clinical effects of water-only fasting in humans. Over the past decade, clinical research into the beneficial and adverse health effects of water-only fasting has progressed with a level of scientific rigor not previously reported [6-10, 26, 36].

Additionally the following existing examples throughout our manuscript address the scientific soundness of prolonged water-only fasting and especially the specific water-only fasting protocol implemented in this study.

Introduction:

Fasting is the partial or complete abstinence of caloric intake for a defined period of time. Research into intermittent fasting, the fasting mimicking diet, and prolonged fasting methods have demonstrated the potential of these therapies to improve overall health and promote immunity and longevity [4]. Prolonged fasting protocols have been shown to improve cardiometabolic markers associated with obesity such as insulin sensitivity, blood lipids, body weight, and abdominal circumference [5-7]. Prolonged fasting is typically conducted as a very-low-calorie or zero-calorie (e.g., water-only fasting) intervention for a period of 2 or more days [8].

Methods:

Medically Supervised, Water-only Fasting & Refeeding Protocol

The medically supervised, water-only fasting and refeeding protocol was implemented by non-research medical personnel at a residential, medical facility according to the facility’s standard protocol as previously reported [8]. Briefly, potential participants were pre-screened before arrival and, if conditionally approved to water-only fast, were instructed to eat a diet consisting of fresh fruits and raw or steamed vegetables for two days prior to initiating the fast. Participants were instructed to consume a minimum of 40 ounces of distilled water per day and limit physical activity during the course of the fast. While fasting, participants remained on-site and non-research, medical personnel monitored vital signs and symptoms twice daily along with weekly serology and urine analysis to monitor electrolyte balance and other physiological functions, such as kidney and liver function. Adverse events were continuously monitored and if necessary the fast was temporarily interrupted with vegetable broth or juice or suspended by initiating the standard refeeding protocol. At the EOF, the fast was broken with a refeeding process consisting of five phases of gradual food introduction, with one phase for every 7-10 days of fasting. Phase one is a mixture of fruit and vegetable juice; phase two includes the addition of raw fruits and vegetables; phase three includes the addition of steamed vegetables; phase four includes the addition of whole grains; and phase five includes the addition of legumes, until participants are eating an exclusively SOS-Free Diet. Only phase one limited daily calorie consumption, and each new phase consisted of a continuation of items from the previous phase with the addition of more complex foods. Refeeding length was at least half of the fasting length. Additionally, participants’ vital signs and symptoms were monitored twice daily for the duration of the treatment, which included fasting and refeeding phases. Approved fasting lengths varied but were no less than 10 days followed by a refeeding period of at least five days.

Conclusion:

Metabolic diseases, particularly CVD and T2D, are a global public health concern and there is increased interest in developing and implementing cost-effective methods of prevention and reversal. The pathophysiology of metabolic disease is complex, and there is variability in how obesity impacts insulin resistance and lipid metabolism, as some obese people remain insulin sensitive without associated cardiometabolic disorders [26, 28]. Nevertheless, early intervention with treatments that result in sustained improvements in biomarkers, such as BW, that correlate with increased cardiometabolic disease risk may prevent the development of these chronic conditions and improve overall health. Water-only fasting improves obesity as well as other markers of CVD risk including lipid profile, FLI, and hsCRP, and the results are sustained for at least six weeks with imperfect adherence to an exclusively whole-plant-food diet.

  1. Was the study approved by Ethical committee or board? Was written consent obtained from the patients to participate in the research study?

Yes. This was clearly indicated in our manuscript’s Institutional Review Board and Informed Consent Statements (see below). For additional clarification, we have added an ethics statement to the beginning of the Methods section (see below).

Institutional Review Board Statement: The study was conducted according to the guidelines of the Declaration of Helsinki, and approved by the Institutional Review Board of the TrueNorth Health Foundation (TNHF-2020-2VAT, April 2, 2020).

Informed Consent Statement:  Informed consent was obtained from all subjects involved in the study.

Ethical Statement

This study (NCT04514146) was conducted according to the guidelines of the Declaration of Helsinki, and approved by the Institutional Review Board of the TrueNorth Health Foundation (TNHF-2020-2VAT).

Reviewer 3 Report

In this study authors studied the implications of water only fasting on cardiometabolic risk factors. It is an interesting study and I congratulate the authors for their work. I have following concerns

1. Is IRB clearance taken
2. How were participants kept motivated to go on with the fasting
3. Please report nausea and gastritis as potential issues of this method as 3 to 4 pt had it (10%)
4. Was Na and Creatinine, Po4 normal at the beginning and the end of the fasting

Author Response

In this study authors studied the implications of water only fasting on cardiometabolic risk factors. It is an interesting study and I congratulate the authors for their work. I have following concerns

  1. Is IRB clearance taken

Yes. This was clearly indicated in our manuscript’s Institutional Review Board and Informed Consent Statements. For additional clarification, we have added an ethics statement to the beginning of the Methods section as well (see below).

Ethical Statement

This study (NCT04514146) was conducted according to the guidelines of the Declaration of Helsinki, and approved by the Institutional Review Board of the TrueNorth Health Foundation (TNHF-2020-2VAT).

  1. How were participants kept motivated to go on with the fasting

This study was conducted as an observational study and the participants were recruited from patients of a residential medical fasting center who were already planning on water-only fasting the minimum fast length before they were contacted by research personnel. The participants were also approved to water-only fast the minimum fast length by non-research medical personnel before they were enrolled in the study.

The research protocol did not interfere with the water-only fasting intervention (i.e., length, monitoring, etc.) or with patient motivation in any way. As such, it is assumed that the participants were self-motivated as they paid for their water-only fasting treatment and only received modest compensation for participation (which they received whether or not they completed the water-only fast, and was approved by the overseeing IRB).

  1. Please report nausea and gastritis as potential issues of this method as 3 to 4 pt had it (10%)

Nausea and gastritis are known complications of fasting as has been previously reported in a retrospective study where safety was assessed [8]. Participants who self-elected to undergo water-only fasting met with a non-research clinician prior to initiating the fast and were subsequently monitored twice-daily by non-research medical personnel at an in-patient facility.

The following was added for clarification in the methods and discussion sections:

Methods

While fasting, participants remained on-site and vital signs and symptoms were monitored twice daily by non-research, medical personnel, along with weekly serology and urine analysis to monitor electrolyte balance and other physiological functions, such as kidney and liver function. Adverse events were continuously monitored and if necessary the fast was temporarily interrupted with vegetable broth or juice or suspended by initiating the standard refeeding protocol.

Discussion

Nonetheless, there is still a concern about the safety and practicality of prolonged water-only fasting in humans [4]. In this study, 95% (38/40) of the enrolled participants were able to complete at least 10 days of fasting with only mild (grade 1) to moderate (grade 2) adverse events. This suggests that - at least in a well-selected population - fasting is well-tolerated. Furthermore, modifications to water-only fasting, such as the inclusion of vegetable broth and/or juice or the temporary suspension of fasting with easily digestible foods, are commonly implemented as part of clinical practice to ensure patient safety, comfort, and well-being and do not appear to have a negative impact on treatment outcomes [6, 8].

  1. Was Na and Creatinine, Po4 normal at the beginning and the end of the fasting?

As mentioned, this was an observational study and the water-only fasting intervention was overseen by trained medical personnel separately from the research study. The study protocol permits access to the weekly serological data and these values were all within normal range at baseline. If they were not this would be considered a contraindication to fasting and the patients would not be approved to fast and thus would not have been enrolled as participants. Serology is not typically run at the end of fasting or during refeeding unless if clinically indicated due to symptoms or previous serology. During the fast, all participants had sodium, creatinine, and potassium within normal range. The water-only fasting protocol implemented at the center was published in a retrospective study which assessed for safety and reported on adverse events occurring in greater than 700 patients who water-only fasted between 2 and 40 days [8]. The study found that the vast majority of adverse events (95%) were mild to moderate in nature and that the risk of having a serious adverse event was less than .05%.We have also added the following to the methods section to clarify(see below).

While fasting, participants remained on-site and non-research medical personnel monitored vital signs and symptoms twice daily along with weekly serology and urine analysis to monitor electrolyte balance and other physiological functions, such as kidney and liver function. Adverse events were continuously monitored and if necessary the fast was temporarily interrupted with vegetable broth or juice or suspended by initiating the standard refeeding protocol.